# The relationship between voting restrictions and COVID-19 case and mortality rates between US counties

Roman Pabayo[1]*, Erin Grinshteyn[2], Brian Steele[1], Daniel M. Cook[3], Peter Muennig[4], Sze Yan Liu[5]

1 University of Alberta School of Public Health, Edmonton, Alberta, Canada, 2 Health Professions Department, University of San Francisco, San Francisco, California, United States of America, 3 School of Public Health, University of Nevada, Reno, Reno, Nevada, United States of America, 4 Columbia Mailman School of Public Health, New York City, New York, United States of America, 5 Public Health Department, Montclair State University, Montclair, New Jersey, United States of America

* pabayo@ualberta.ca

## Abstract

### Background

Since the 2010 election, the number of laws in the U.S. that create barriers to voting has increased dramatically. These laws may have spillover effects on population health by creating a disconnect between voter preferences and political representation, thereby limiting protective public health policies and funding. We examine whether voting restrictions are associated with county-level COVID-19 case and mortality rates.

### Methods

To obtain information on restricted access to voting, we used the Cost of Voting Index (COVI), a state-level measure of barriers to voting during a U.S. election from 1996 to 2016. COVID-19 case and mortality rates were obtained from the New York Times' GitHub database (a compilation from multiple academic sources). Multilevel modeling was used to determine whether restrictive voting laws were associated with county-level COVID-19 case and mortality rates after controlling for county-level characteristics from the County Health Rankings. We tested whether associations were heterogeneous across racial and socioeconomic groups.

### Results

A significant association was observed between increasing voting restrictions and COVID-19 case (ß = 580.5, 95% CI = 3.9, 1157.2) and mortality rates (ß = 16.5, 95% CI = 0.33,32.6) when confounders were included.

### Conclusions

Restrictive voting laws were associated with higher COVID-19 case and mortality rates.

**Data Availability Statement:** The data underlying the results presented in the study are available from https://github.com/nytimes/covid-19-data.

**Funding:** This study was supported by the Canada Research Chair program, of which RP is a recipient of. The funders had no role in study design, data collection and analysis, decision to publish, or preparation of the manuscript.

**Competing interests:** The authors have declared that no competing interests exist.

## Introduction

The right to vote offers marginalized communities the opportunity to participate in the political process, thereby enjoying a policy environment that better represents their needs. This potentially leads to beneficial population health outcomes [1–3]. For example, voters who believe in access to Medicaid but confront barriers to access to voting may not have their voices heard in the electoral process, and therefore not have access to health services. Differences in voter participation due to social and economic inequities can significantly affect electoral outcomes [1]. To the extent that voter restrictions are regional, they may explain regional differences in the case and death rates due to COVID-19 in the U.S.

During the first year of the COVID-19 pandemic, almost 30 million Americans had documented COVID-19 infections, a case rate of 9,021 per 100,000. Over 540,000 died from these infections, resulting in a death rate of 163 deaths per 100,000 [4]. The spatial distribution of COVID-19 cases and deaths suggest that geographical and contextual attributes have contributed to its spread [5, 6].

One plausible contributor to the variation in COVID-19 incidence rates across geographic regions is the variation in the implementation or enforcement of essential public health measures to prevent the spread of COVID-19 [7]. Governments face competing resource and policy demands, and electorate preferences may decide which policies are prioritized and which are not. When a portion of the electorate is restricted from voting, there exists the possibility that fewer voices are heard resulting in tangible population health harms. For example, those who see mask mandates as harming indoor commerce may have a stronger political voice than those who oppose them. Such restrictions likely contributed to variations in infection and mortality rates [8]. It is difficult for public health agencies to prevent the transmission of COVID-19 when local and state elected officials oppose the policies that public health officials recommend.

A second contributor to geographical variation is funding for local public health agencies. Elected officials who oppose measures to prevent the spread of COVID-19 are also less likely to provide adequate funding to local public health agencies that are charged with making recommendations or implementing regulations to prevent the spread of COVID-19. With less funding, such agencies are also less likely to be able to oversee testing, contact tracing, and isolation, as well as vaccination programs; critical public health interventions for preventing the spread of COVID-19 [9].

A final contributor to geographical variation is funding for the social determinants of health. Anti-poverty policies, such as Medicaid expansion and financial aid, are designed to reduce material hardship and improve the health of socio-economically-disadvantaged populations. Poverty is associated with a larger burden of disease than smoking and obesity combined [10]. Yet voter restrictions, which have a disproportionate effect on individuals from lower socioeconomic backgrounds and people of color, may produce a shift from candidates who are more likely to support such policies that address such socioeconomic inequities compared to those who are less likely to support them. People from such socio-demographic groups are also more likely to work in essential services and businesses that are at higher risk for COVID-19 infection since they cannot work from home and are more likely to be in workplaces where social distancing is not possible [11–13]. These essential workers would also be disproportionately affected by restrictive voting laws, which may lead to elected officials who discount public health measures.

Voter restrictions tend to target the same socio-demographic groups who are most likely to contract or die from COVID-19 [14, 15]. In fact, current voting restrictions may be viewed as "Jim Crow" laws 2.0 as they indirectly target Black-Americans [16]. For example, strict ID laws

in Texas have been shown to keep Black and Hispanic voters from casting ballots despite their desire to vote [17]. Another study of polling locations in Los Angeles found that barriers to voting were highest in lower-income neighborhoods or those with a higher proportion of racial/ethnic minorities [18]. Furthermore, these same populations may be more likely to support elected officials who favor public health and social welfare policies. Such policies have been shown in a meta-analysis to improve health [19]. Voting infrastructure—including voter registration processes, polling location, ease of access, early voting, remote voting, the ability to cast a provisional ballot, inconvenient polling place hours, and administrative capacity— have tangible impacts on voter participation [20–22]. Thus, areas with larger barriers to voting may expect to see a reduction in political participation.

However, there is limited research on the association between voting restrictions and health inequities [3]. Since laws limiting voting rights tend to target groups most at risk of COVID-19 infection and mortality, it is important to control for such risk factors. Sociodemographic risk factors that are common for chronic illness—race, household income, and occupation—also place individuals at higher risk for COVID-19 infection. Once infected, members of these groups are more vulnerable to severe morbidity and mortality [23]. Furthermore, risk factors for COVID-19 are distinctive and reflective of public health policies.

In light of recent increases in legislation to restrict voting, we investigated the relationship between voting restrictions and county-level COVID-19 case and mortality rates in the US. The purpose of this investigation is to determine whether restrictive voting laws are associated with higher COVID-19 case and mortality rates and whether this relationship was heterogeneous across racial and socioeconomic groups.

## Methods

### Sources of data

Data from 3142 counties within the 50 states and Washington, DC were obtained for this eco-logical study. County-level COVID-19 data were obtained via the New York Times' GitHub database [4]. The County Health Rankings compiles sociodemographic data on US counties [24]. Use of the aggregated data required no ethical review beyond that already done for creation of the database from which it was extracted.

### Measures

The main exposure of interest is access to voting measured using the Cost of Voting Index (COVI). COVI is a global measure of difficulty in voting during an election [20] that uses a principal components analysis on 33 different state election laws from 1996 to 2016. Examples of barriers included are the number of days before an election that registrations must occur; whether felons are allowed to register; whether pre-registration is allowed; whether a photo ID is required; and the number of hours the polls are open. The higher the score, the more difficult it is to vote [20]. The construct validity of the COVI scale has been tested, and voter turn-out is lower in states with higher index values [20].

Political partisanship is a potential confounder in the association between voting restrictions and COVID cases because it can be associated with both voting restrictions and COVID-19 outcomes. For example, local governments and national political parties sometimes politicized public health responses to the pandemic. States that lean more toward the Republican party, typically those with more voting restrictions, also implemented fewer COVID-19 mitigation strategies and were slower to issue policy responses compared to states that lean more toward the Democrat party [25]. On the individual level, partisan political affiliation is a very strong predictor of willingness to adhere to social distancing and other efforts to 'stop the

spread' [26, 27]. Thus, political partisanship was measured by the proportion in each county who voted for Donald Trump in the 2016 US election. Proportions ranged from 4.3% to 100.0%, with an average of 66.7% (SD = 16.2%). The proportion of voters who voted for Donald Trump was tested as a covariate, an independent predictor, and was used as an interaction term with COVI.

Other county characteristics included as covariates included population size, median household income, proportion Black, proportion rural, proportion under the age of 18, and proportion over the age of 65 years.

## Outcome measures

We used state and County-level cumulative COVID-19 cases and mortality rates from January 20, 2020 to March 19, 2021 as primary outcome measures for this investigation.

## Statistical analysis

We first determined the bivariate associations between COVI and state-level COVID-19 case and mortality rates. Scatter plots were also created to visualize these bivariate relationships.

Since US counties were nested within states, we conducted multilevel linear regression to investigate the relationship between COVI and COVID-19 case rates (per 100,000) and mortality rates (per 100,000). We first estimated a state-level intercept-only model to calculate the IntraClass Correlation (ICC), representing the degree of variability of case and mortality rates between US states. For example, the proportion of variance of each outcome explained by the county- and state- levels can be computed using the ICC. Second, we measured the unadjusted association between the COVI index and each outcome. Third, we added county-level characteristics into the models. Fourth, we tested COVI-proportion Black and COVI-median income interaction terms to determine if the associations between COVI index and COVID outcomes were heterogeneous across sociodemographic groups. Finally, we added a COVI x proportion Trump voters interaction term. All analyses were conducted using Stata v. 14.0.

Since voting restrictions within a state may be a marker of political partisanship, we first determined the correlation between COVI and proportion Trump voters. The Pearson correlation coefficient was 0.12. We conducted sensitivity analyses in which proportion Trump voters was included and excluded from the final model specification to ascertain the extent to which there is confounding by political partisanship.

## Results

US County characteristics are found in Table 1. The average proportion Black was 9.0 (SD = 14.3) and the median income was 52,767.90 USD (SD = 13,865.82). The average COVID-19 case rate was 9,311.52 per 100,000 (SD = 2,985.4) and ranged from 260.6 to

**Table 1. Characteristics of the US Counties.**

| County Level Characteristics | Mean (SD) | Range |
|---|---|---|
| Proportion Trump Voters, % | 66.7 (0.16) | 4.3 to 100 |
| Population | 104,372 (358,186) | 88 to 10,105,518 |
| Median income, USD | 52768(13,865.82) | 25,385 to 140,382 |
| Proportion Black | 9.0(14.3) | 0 to 85.4% |
| Proportion Rural | 58.6(31.4) | 0 to 100 |
| Proportion <18 years | 22.1(3.5) | 0 to 42.0 |
| Proportion >65 years | 19.3(4.7) | 4.8 to 57.6 |

36,206.9 per 100,000. The average COVID-19 mortality rate was 182.4 per 100,000 (SD = 110.1) and ranged from 0 to 842.3 COVID-19 deaths per 100,000. According to the null model, the ICC for case rate was 0.49 (95% CI = 0.39,0.59) while the ICC for mortality rate was 0.33 (95%CI = 0.25, 0.44). ICC values indicate 49% and 33% of the variance of COVID-19 case rates and COVID-19 mortality rates were explained at the state-level, respectively.

The bivariate associations between COVI and COVID-19 outcomes indicate that a one SD increase in COVI was associated with an increase in state-level COVID-19 case (ß = 256.8, 95% CI = -376.7, 890.3) and mortality rates (ß = 5.6, 95%CI = -7.4,18.7), but these estimates were not statistically significant. The scatterplots of the association between COVI and COVID-19 cumulative incidence rate and mortality rate can be found in S1 Appendix.

The association between COVI and COVID-19 case rates can be found in Table 2. In the unadjusted model, an increase in SD of COVI was associated with an increase in COVID-19 case rates (ß = 785.3, 95% CI = 165.0, 1405.5). The association remained significant when

**Table 2. The relationship between Cost of Voting Index and COVID-19 case rate during the first year of the pandemic.**

| | Crude Relationship | Adjusted Relationship | Adjusted + % Trump Supporters | Adjusted + Black Interaction | Adjusted + Black Interaction and % Trump Voters Interaction | Adjusted + Median Income Interaction | Adjusted + %Trump Voters Interaction + Median Income Interaction |
|---|---|---|---|---|---|---|---|
| | β | β | β | β | β | β | β |
| | 95% CI | 95% CI | 95% CI | 95% CI | 95% CI | 95% CI | 95% CI |
| Intercept | 8613.5 | 8,781.5 | 8932.0 | 8,997.40 | 9035.9 | 8,906.40 | 8910.6 |
| | (7,992.3, 9,234.6) | (8204.4,9358.7) | (8382.6,9481.5) | (8445.0, 9549.8) | (8484.1, 9587.8) | (8360.7, 9452.1) | (367.4, 9453.8) |
| Cost of Voting Index (COVI) Z-score | 785.3 | 580.5 | 465.4 | 401.2 | 356.4 | 489.1 | 480.1 |
| | (165.0,1405.5) | (3.9,1157.2) | (-78.1, 1008.9) | (-145.2, 947.5) | (-189.9, 902.7) | (-50.5, 1028.7) | (-57.1, 1017.3) |
| **County-Characteristics** | | | | | | | |
| Proportion Trump Voters | | | 434.6 | 433 | 447.2 | 449.7 | 462.7 |
| | | | (299.7, 569.4) | (298.3, 567.8) | (311.9, 582.6) | (314.3, 585.2) | (326.2, 599.2) |
| Population, Z-Score | | 7.9 | 33.0 | 21.9 | 24.8 | 29.7 | 33.9 |
| | | (-79.7,95.4) | (-64.5,130.5) | (-75.9, 119.7) | (-73.0, 122.6) | (-67.8, 127.1) | (-63.7, 131.5) |
| Median income, Z-Score | | -730.7 | -708.6 | -705.6 | -693.5 | -688.8 | -677.5 |
| | | (-834.8, -626.7) | (-812.1, -605.0) | (-809.0, -602.1) | (-797.5, -589.4) | (-793.9, -583.7) | (-783.7, -571.4) |
| Proportion Black, Z-Score | | -111.5 | 158.7602 | 326.7 | 349 | 157.1 | 128.2 |
| | | (-234.0,11.0) | (11.4,306.2) | (126.7, 526.6) | (148.1, 549.9) | (9.8, 304.4) | (-24.0, 280.5) |
| COVI X Proportion Black Interaction Term | | | | -208.2 | -288.2 | | |
| | | | | (-375.9, -40.6) | (-472.6, -103.9) | | |
| COVI X Proportion Trump Interaction Term | | | | | -103.0 | | -69.8 |
| | | | | | (-202.1, -4.0) | | (-163.2, 23.7) |
| COVI X Median Income Interaction Term | | | | | | -87.2 | -104.1 |
| | | | | | | (-167.9, -6.5) | (-187.8, -20.3) |
| Proportion Rural, Z-Score | | -261.2 | -381.2 | -376.2 | -365.7 | -388.2 | -383.7 |
| | | (-2366.6, -155.9) | (-491.2, -271.2) | (-486.1, -266.2) | (-476.0, -255.3) | (-498.3, -278.1) | (-494.0, -273.5) |
| Proportion <18 years | | 148.3 | 108.6 | 115.2 | 103 | 119 | 111 |
| | | (33.6,263.0) | (-9.1, 226.4) | (-2.6, 233.0) | (-15.3, 221.3) | (0.9, 237.1) | (-7.5, 229.6) |
| Proportion >65 years | | -774.9 | -808.9 | -799.8 | -806.5 | -801.1 | -806.5 |
| | | (-899.4, -650.4) | (-934.8, -683.0) | (-925.8, -673.8) | (-932.6, -680.4) | (-927.2, -675.1) | (-932.8, -680.3) |

**Table 3. The relationship between Cost of Voting Index and COVID-19 death rate during the first year of the pandemic.**

| | Crude Relationship | Adjusted Relationship | Adjusted + % Trump Voters | Adjusted + Black Interaction | Adjusted + Proportion Black Interaction and % Trump Voters Interaction | Adjusted + Median Income Interaction | Adjusted + % Trump Voter Interaction + Median Income Interaction |
|---|---|---|---|---|---|---|---|
| | β | β | β | β | β | β | β |
| | 95% CI | 95% CI | 95% CI | 95% CI | 95% CI | 95% CI | 95% CI |
| Intercept | 165.0 | 174.4 | 177.3 | 178.2 | 177.2 | 174.2 | 174.20 |
| | (147.6,182.3) | (158.2,190.7) | (161.1, 193.6) | (161.9, 194.5) | (160.9, 193.6) | (158.4, 189.9) | (158.5, 189.9) |
| Cost of Voting Index (COVI) Z-score | 27.3 | 16.5 | 15 | 14.2 | 15.2 | 17.70 | 17.7 |
| | (10.1,44.5) | (0.33,32.6) | (-1.0, 31.0) | (-1.9, 30.3) | (-0.9, 31.4) | (2.2, 33.2) | (2.2, 33.2) |
| **County-Characteristics** | | | | | | | |
| % Trump Voters | | | 4.8 | 4.8 | 4.5 | 6.6 | 6.6 |
| | | | (-0.5, 10.2) | (-0.49, 10.1) | (-0.8, 9.8) | (1.3, 11.9) | (1.3, 12.0) |
| Population, Z-Score | | 4.0 | 3 | 2.8 | 2.7 | 2.6 | 2.6 |
| | | (0.6,7.5) | (-0.9, 6.8) | (-1.1, 6.7) | (-1.1, 6.6) | (-1.3, 6.4) | (-1.3, 6.4) |
| Median income, Z-Score | | -31.9 | -31.7 | -31.6 | -31.9 | -29.4 | -29.3 |
| | | (-35.9, -27.8) | (-35.8, -27.6) | (-35.7, -27.5) | (-36.1, -27.8) | (-33.5, -25.2) | (-33.5, -25.1) |
| Proportion Black, Z-Score | | 9.9 | 12.8 | 14.9 | 14.4 | 12.6 | 12.4 |
| | | (5.1,14.6) | (7.0, 18.6) | (7.1, 22.8) | (6.5, 22.3) | (6.9, 18.4) | (6.5, 18.4) |
| COVI X Proportion Black Interaction Term | | | | -2.6 | -0.7 | | |
| | | | | (-9.2, 4.0) | (-7.9, 6.5) | | |
| COVI X Trump Interaction Term | | | | | 2.5 | | -0.4 |
| | | | | | (-1.4, 6.4) | | (-4.1, 3.2) |
| COVI X Median Income Interaction Term | | | | | | -10.2 | -10.3 |
| | | | | | | (-13.3, -7.0) | (-13.6, -7.0) |
| Proportion Rural, Z-Score | | -4.3 | -5.9 | 23.9 | -6.1 | -6.8 | -6.7 |
| | | (-8.4, -0.2) | (-10.3, -1.6) | (-10.2, -1.5) | (-10.5, -1.7) | (-11.1, -2.4) | (-11.1, -2.4) |
| Proportion <18 years | | 23.1 | 23.8 | 19.6 | 24.2 | 25 | 24.9 |
| | | (18.6,27.6) | (19.1, 28.5) | (19.2, 28.6) | (19.5, 28.9) | (20.3, 29.6) | (20.2, 29.6) |
| Proportion >65 years | | 19.5 | 19.5 | 178.2 | 19.8 | 20.4 | 20.4 |
| | | (14.6, 24.3) | (14.5, 24.5) | (14.6, 24.6) | (14.8, 24.8) | (15.4, 25.4) | (15.4, 25.3) |

adjusted for county characteristics (ß = 580.5, 95% CI = 3.9, 1157.2). When we included the proportion Trump voters to account for partisanship, the association between COVI and case rates remained, but was no longer statistically significant (ß = 465.4, 95% CI = -78.1, 1008.9). However, an SD increase in proportion Trump voters was associated with a significant increase in case rates (ß = 434.6, 95% CI = 299.7,569.4).

Table 3 highlights the results between voting restrictions and COVID-19 mortality rates. The crude relationship indicated an SD increase in COVI was related to a significant increase in mortality rate (ß = 27.3, 95% CI = 10.1, 44.5). Findings remained significant when controlling for confounders (ß = 16.5, 95% CI = 0.33,32.6). Furthermore, when we included proportion Trump voters, a one SD increase in COVI was associated with an increase in mortality rates (ß = 15.0, 95% CI = -1.0,31.0).

The COVI-proportion Black interaction terms were significant when COVID-19 case rate (ß = -217.9, 95% CI = -387.8, -48.0) was the outcome but not when the mortality rate was the outcome (ß = -2.7, 95% CI = -9.3, 3.8). Fig 1 depicts the relationship between COVI and both

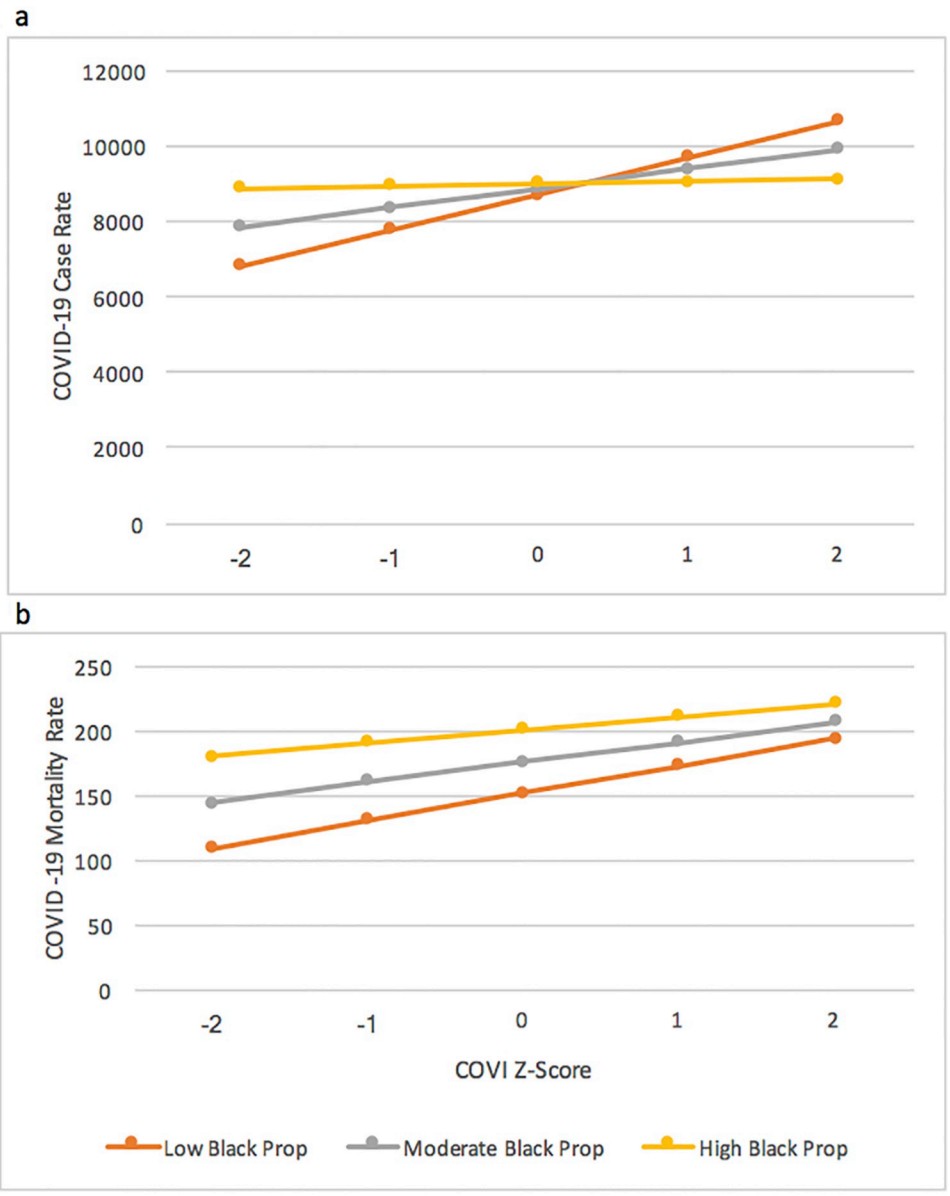

**Fig 1. The relationship between Cost of Voting Index (COVI) z-score and COVID-19 outcomes across proportion of US county that is Black (n = 3,106 counties).** a. The adjusted relationship between Cost of Voting Index (COVI) score and county COVID case rate by proportion Black. b. The adjusted relationship between Cost of Voting Index (COVI) score and county COVID mortality rate by proportion Black.

outcomes by proportion Black. COVI is associated with an increase in case rates within counties with low proportions of Black-Americans. Although increasing COVI was related to higher mortality rates, the relationship was homogenous across differing proportions of Black populations within counties. However, the mortality rates were higher within counties with high proportions of Black residents than counties with moderate or low proportions of Black residents, regardless of the degree of restrictions. The COVI-proportion Black interaction term remained significant (ß = -288.2, 95% CI = -472, -103.9) when an interaction term for COVI and the proportion of Trump voters was included in the model.

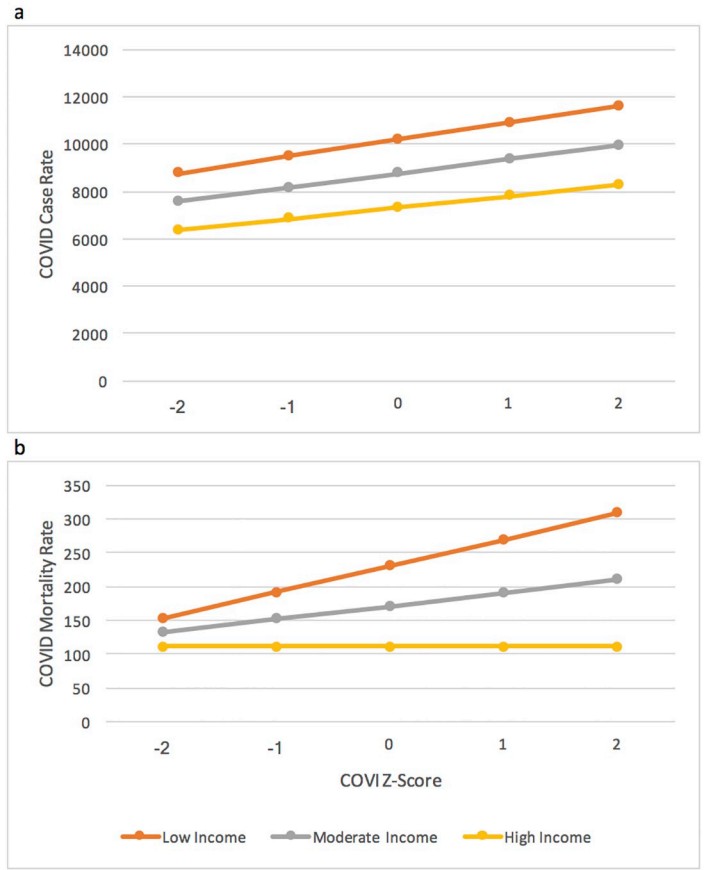

**Fig 2. The relationship between Cost of Voting Index (COVI) z-score and COVID-19 outcomes across US county median income (n = 3,106 counties).** a. The adjusted relationship between Cost of Voting Index score and county COVID case rate by county median income. b. The adjusted relationship between COVI score and county COVID mortality rate by county median income.

The COVI-median income interaction term was significant when the COVID-19 mortality rate was the outcome. As shown by Fig 2, an increase in SD was significantly related to higher mortality rates among counties with low median household income. However, it was constant among counties with higher median household incomes. Furthermore, mortality rates were higher within counties with low median household incomes than in counties with moderate and high median household incomes. Thus, the relationship between COVI and COVID-19 case rates was not heterogeneous across levels of median household incomes. The COVI median income interaction (ß = -10.3, 95% CI = -13.6, -7.0) term remained significant when COVI-proportion Trump voters interaction term was included in the model.

## Discussion

We observed a significant relationship between increasing voting restrictions and COVID-19 case and mortality rates. Furthermore, this relationship proved to be heterogeneous across socio-demographic groups. Counties with lower median incomes experienced higher COVID-19 mortality rates when there were numerous voting restrictions, but less so when fewer restrictions were in place. Further research is needed to identify plausible mechanisms for this spillover effect. Candidate hypotheses include reductions in funding for public health agencies,

reductions of funding social determinants of health, and/or by limiting jurisdictions from implementing public health measures, such as social distancing and masking, which should have helped to prevent the spread of COVID-19.

We observe a significant relationship between voting restrictions within a US state and cumulative COVID-19 case and mortality rates during the first year of the pandemic. Voting restrictions were not associated with case rates within communities with high proportions of Black residents. However, they were associated with higher case rates within counties with low and moderate proportions of Black populations. This heterogeneity in case rates may imply differences in baseline social conditions in these counties or high levels of local political organization. Communities with high proportions of Black residents had consistently high case rates, suggesting these communities had individual and community-level risk factors that placed them at high risk for COVID infection regardless of public health interventions (i.e., poverty, over-crowding, or high levels of chronic diseases). However, the variable indicating the proportion of Black residents was an effect modifier for county-level mortality. This suggests that improving access to medical services may contribute to lower mortality due to COVID-19.

Laws that make it difficult for minorities, particularly Black Americans, to vote are examples of structural racism [28]. The Voting Rights Act of 1965 prohibits racial discrimination in elections in the US. The Voting Rights Act contained a "preclearance" requirement, which prohibited jurisdictions from implementing any change affecting voting without receiving pre-approval from the US Attorney General or the US District Court for D.C. This was done to ensure that any change in the law does not discriminate against protected minorities [29]. However, the Supreme Court's 2013 landmark decision, Shelby County v. Holder, ruled that the preclearance clause in the Voting Rights Act was unconstitutional. Thus, jurisdictions have become able to easily implement restrictive voting laws that disproportionately affect racialized minorities and those from low socioeconomic groups since 2013.

Fourteen US states legislated restrictive voting laws after Shelby County v. Holder in time for the 2016 Presidential election [30]. Of the 11 states with the highest African-American turnout in 2008, six had legislated new voting restrictions for the 2016 election [30]. Most recently, as of March 24, 2021, state governments of 47 states have introduced 361 restrictive voting bills [31]. Thus, voting restrictive laws have gained momentum and can influence election outcomes, impacting population health and widening health inequities.

When individuals are prevented from voting, those elected are not accountable to all of their constituents' interests. High costs to voting reduce turnout, particularly among people from low socioeconomic status groups and racialized minorities [20, 21, 28]. When voter turnout is higher, i.e., people from low socioeconomic status and minorities participate in elections, the effects on electoral outcomes may be significant [1].

For example, increased access to life-saving goods, such as health insurance, may result. States with high proportions of Black-Americans who support Medicaid expansion tend to be represented by elected officials who oppose Medicaid expansion [32]. The same may be true of support for other government welfare policies that support the social determinants of health. The social determinants of health, such as income or education, account for a larger disease burden than traditional risk factors, such as smoking or obesity [10]. Government policies that address the social determinants of health have been shown in a meta-analysis of randomized-controlled trials to improve health and reduce mortality [19]. In the same vein, those representatives may be against other life-saving policies, particularly those needed during a pandemic of an infectious disease including public health mitigation strategies such as mask-wearing, social distancing, lockdowns involving the closure of schools and businesses, and staying home. In addition to these measures, governments can provide economic support for their

residents so they do not have to work outside of their homes, which would decrease the risk for COVID-19 infection. For essential services, governments can support initiatives that provide personal protective equipment and adequate ventilation.

In addition to the main effect of voting restrictions on COVID-19 mortality rates, our results suggest heterogeneity by median household income. Voting restrictions were associated with increased COVID-19 mortality among counties with low median household income, which indicates that lower income populations experience a larger burden of the pandemic, and those who have faced barriers to voting experienced higher mortality rates. This result points to an often-over-looked mechanism in which populations living in poverty or low socioeconomic status leads to adverse health outcomes. Those from lower socioeconomic backgrounds may be blocked from participating in elections by voting restriction laws. For example, one in ten Americans does not have a government-issued photo ID [33]. In addition to students, African-Americans, and Latinos, those from low socioeconomic backgrounds are less likely to have photo ID [33]. Policies affecting administrative capacity may also adversely affect lower-income populations. Median census tract household income is positively associated with the number of election judges at polling locations and negatively associated with the number of people waiting in line to vote at 7 PM and overall wait times, both of which result from local policies dictating election procedures [21]. Qualitative observations at polling sites also found more confusion and increased police attendance in lower income areas [21]. Thus, those from lower socioeconomic backgrounds are less likely to be represented by elected officials who would then enact laws that act in their best interest, which can profoundly impact their health.

Results of this study should be interpreted with caution due to several limitations. First, the study design was an ecological study, limiting our findings to the county and state levels. This limits inferences made at the individual level. Disaggregated data are needed to determine the magnitude of the association between the voting restrictions and COVID-19 illness and mortality risk at the individual level. Secondly, data were not available for case and mortality rates by race. Instead, we determined the relationship between the proportion Black and COVID-19 case and mortality rates. Also, due to data limitations, we could not test specific mechanisms through which voting restrictions lead to increased COVID-19 case and mortality rates, particularly among vulnerable groups.

Participating in voting is associated with beneficial health outcomes at the county level [1–3]. However, this relationship likely suffers from endogeneity. Thus, it is likely that while voting may lead to improved health outcomes, better health outcomes may also be associated with increased voter participation. Further research should utilize longitudinal data to test potential mediators, such as the enactment of policies that could either spread or prevent COVID-19 infection and mortality. Likewise, to the extent that social policy experiments impact health, it would be useful to explore the impact of new social policy experiments on voting behaviors.

Among US counties with moderate median household incomes, voting restrictions were associated with higher COVID-19 death rates. However, voting restrictions were not associated with COVID-19 death rates among counties with high household incomes. These findings indicate that to obtain population health equity, access to voting should not be limited. Improving access could help communities achieve health equity.

## Supporting information

**S1 Appendix. The correlation between COVI and cumulative incidence rate and mortality rate.**
(DOCX)

## Author Contributions

**Conceptualization:** Roman Pabayo, Erin Grinshteyn, Daniel M. Cook, Peter Muennig, Sze Yan Liu.

**Data curation:** Brian Steele.

**Formal analysis:** Roman Pabayo, Sze Yan Liu.

**Investigation:** Roman Pabayo, Erin Grinshteyn, Daniel M. Cook, Peter Muennig.

**Methodology:** Roman Pabayo, Peter Muennig, Sze Yan Liu.

**Supervision:** Roman Pabayo.

**Writing – original draft:** Roman Pabayo, Peter Muennig, Sze Yan Liu.

**Writing – review & editing:** Roman Pabayo, Erin Grinshteyn, Brian Steele, Daniel M. Cook, Peter Muennig, Sze Yan Liu.

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
