## [Decision Letter · Decision Letter 0]

17 Jan 2022

PONE-D-21-34873The Relationship Between Voting Restrictions and COVID-19 Transmission and Mortality Rates within US CountiesPLOS ONE

Dear Dr. Pabayo,

Thank you for submitting your manuscript to PLOS ONE. After careful consideration, we feel that it has merit but does not fully meet PLOS ONE’s publication criteria as it currently stands. Therefore, we invite you to submit a revised version of the manuscript that addresses the points raised during the review process. Please submit your revised manuscript by Mar 03 2022 11:59PM. If you will need more time than this to complete your revisions, please reply to this message or contact the journal office at plosone@plos.org. Please include the following items when submitting your revised manuscript:A rebuttal letter that responds to each point raised by the academic editor and reviewer(s). You should upload this letter as a separate file labeled 'Response to Reviewers'.A marked-up copy of your manuscript that highlights changes made to the original version. You should upload this as a separate file labeled 'Revised Manuscript with Track Changes'.An unmarked version of your revised paper without tracked changes. You should upload this as a separate file labeled 'Manuscript'.

We look forward to receiving your revised manuscript.

Kind regards,

Natalie J. Shook

Academic Editor

PLOS ONE

Journal Requirements:

4. Thank you for stating the following in the Funding Section of your manuscript:

“Roman Pabayo is a Tier II Canada Research Chair in social and health inequities throughout the lifespan”

We note that you have provided additional information within the Funding Section that is not currently declared in your Funding Statement. Please note that funding information should not appear in other areas of your manuscript. We will only publish funding information present in the Funding Statement section of the online submission form.

Reviewers' comments:

Reviewer's Responses to Questions

**Comments to the Author**

1. Is the manuscript technically sound, and do the data support the conclusions?

Reviewer #1: Yes

Reviewer #2: Yes

Reviewer #3: Partly

2. Has the statistical analysis been performed appropriately and rigorously? 

Reviewer #1: I Don't Know

Reviewer #2: Yes

Reviewer #3: Yes

3. Have the authors made all data underlying the findings in their manuscript fully available?

Reviewer #1: Yes

Reviewer #2: Yes

Reviewer #3: Yes

4. Is the manuscript presented in an intelligible fashion and written in standard English?

Reviewer #1: Yes

Reviewer #2: Yes

Reviewer #3: Yes

5. Review Comments to the Author

Reviewer #1: Review of “The Relationship Between Voting Restrictions and COVID-19 Transmission and Mortality Rates within US Counties”

The title contains two misconceptions: 1. “transmission” refers to spread of the virus in time but this is a cross-sectional study of identified cases after one year. 2. the study is not of differences “within counties”. It examines differences among counties.

First sentence: “Participating in voting is related to beneficial health outcomes at the community level [1, 2]”. The references say that fewer people vote in communities where people have more health problems. That does not mean that voting has an effect on health status. People with impaired health may vote less or both could be a result of other factors. It is doubtful that people vote on the basis of concerns for preventive public health. The exception may have been in the 2020 election because of Trump’s lies re: COVID19. The turnout for the 2020 elections was very high by historical standards and probably cost Trump the election (https://link.springer.com/article/10.1007/s00148-020-00820-3). The mortality rate in a county was related to percent Trump voters and testing, corrected for numerous factors associated with interpersonal spread of a highly contagious virus, but that is likely the result of Trump voters’ devotion to Trump’s lies rather than voter suppression which is primarily a state level, not county level, policy. (https://rdcu.be/cCwdm). This study controls for only a few of the potential confounders.

The report repeats the prevalent cliché among many people who study social factors and health, “social determinants of health”. Social and economic factors contribute to increase or decreased risk of certain injuries and illnesses but none are determinative. Also, the effects vary in opposite directions depending on the outcome studied. Social support is related to positive health outcomes for some chronic health problems but social contacts in a pandemic spread the disease.

Voter suppression is a threat to democracy and may result in candidates adverse to public health winning elections but the study reported in this paper does not adequately explicate the issue.

Reviewer #2: Thank you for the opportunity to review this manuscript.

Overall, I think the authors make an important contribution to scholarship at the nexus between public health and political science. The authors' argument is as novel as it is plausible. The theory section does a good job of establishing a link between anti-democratic reforms and public health outcomes while doing a good job of citing the relevant political science literature. The methodological approach strikes me as pragmatic. In an ideal world, a county-level indicator for cost of voting would have been preferable to the state-level measure that was used here. Yet, the authors did the best that they could with the data that was available to them. Multilevel models tend to be overused; here, it was more than appropriate to opt for such models. All of this translates into an original, nifty contribution that will be well cited. Thus, my recommendation is that this paper be published with the following (minor) revisions.

1. My first suggestion relates to how partisanship is addressed in the methods section. The reason the authors are controlling for states' partisan leanings is that the relationship of interest could be driven by this third factor. I suggest that the authors make this clearer in two ways. First, for consistency, they should remove any mention of "ideology," as the concept they are interested in is rather "partisanship." Second, I would be more explicit about why partisanship is a crucial control to have: this variable influences *both* voting restrictions and COVID-19 rates. In other words, partisanship is a confounder that meets the back-door criterion.

2. A second suggestion would be to make the figures more intelligible. Figures 1a and 1b lack axis labels, and the titles are not at all clear. It was initially hard to know what I was looking at. What is the horizontal axis, and why does it go from 1 to 5? What are the vertical axes, and why are they different in each figure? I would like to see a detailed legend explaining all of this very clearly. Relatedly, why are there to confidence intervals surrounding the point estimates? This would help readers assess visually whether the plotted heterogeneous relationships meet standard levels of statistical significance. Finally, what is the total number of cases (i.e., unique observations)? I cannot find this information at the bottom of either regression table. Is it 3142 observations (one per county), or are there repeated county observations?

3. A third suggestion would be to present (or at least mention in the text) some additional *bivariate* descriptive findings. Which states are the best and worst in terms of voting, and how do these states rank in terms of COVID-19 cases and deaths? Are some of the hardest-hit counties in states figuring among the worst culprits? These are only suggestions. I am confident the authors will be able to include one or two such data snippets to illustrate their paper's main argument.

4. To me, the paper's main weakness is that it uses a state-level indicator for the explanatory variable whereas the dependent variable is measured at the county level. As an additional assurance that the results are robust despite the different levels of observation, I would like to see reported correlation coefficients between COVI and each of the two COVID-19 outcomes *at the state level*. Put differently, I would like the authors to aggregate their county-level COVID-19 indicators to see if there is a bivariate association between the explanatory variable and each dependent variable at the state level. The authors might also find it worthwhile to include scatterplots showing the raw, bivariate relationship between their explanatory and dependent variables.

5. Additionally, the paragraph in page 4 starting by "A final contributor to geographical variation is..." should be revised for clarity, as the main substantive idea was not easy to follow. One sentence, in particular, seems to bundle together COVID-19 and voter restrictions in explaining the election of politicians who are averse to public health. I do not understand this point. Are the authors arguing that COVID-19 cases and deaths depress support for those politicians who are more likely to support measures to stop the spread of the virus? I am not convinced that COVID-19 exposure changed how people voted (see Mendoza Avina & Sevi, 2021). If anything, COVID-19 shifted the electorate towards politicians more willing to embrace sound public health measures (i.e., Democrats; see Warshaw, Vavreck, & Baxter-King, 2020).

6. Finally, I do not know whether the authors did this intentionally or not, but I noticed that they tend to avoid using the terms "Democratic" and "Republican" to describe those politicians who tend to support or oppose certain policies. For example, they argue that nonwhite voters and those from socioeconomic disadvantaged backgrounds are more likely to support politicians who support public health measures to slow the spread of the virus. This is technically true, but a simpler and more accurate characterization of the political dynamics at hand is that minority and poor voters support the Democratic Party, which fully supports the policies in question (unlike the Republican Party). The most influential theories of vote choice and preference formation tell us that most voters will support or oppose candidates primarily because of the party to which they belong, not the policies they promote. Voters follow their preferred party's issue positions, meaning that most of them would vote the same way regardless of their respective party's public health platform. Thus I would encourage the authors to characterize citizens' electoral behavior in terms of partisanship rather than public policy.

Reviewer #3: The authors test whether voting restrictions are associated with COVID-19 transmission and death rates, and whether the nature of this association differs as a function of (1) the proportion of Black residents, and (2) median household income.

There is a lot to like about this work. This paper poses a (to my knowledge) novel and important question, and tests it in a relatively compelling fashion. The introduction is well crafted and concise, and makes a logical case for the proposed associations.

My primary concern is that the authors may not have adequately controlled for the possible confounding influence of Republicanism. That is - as the authors note - the pandemic was highly politicized, with Republican politicians tending to impose fewer restrictions. At the same time, voting restrictions (e.g., voter ID laws) are also more common in Republican-leaning areas. Thus, it is possible that any observed associations between voting restrictions and covid transmission / death rates are due to greater Republicanism, rather than the voting restrictions per se.

The authors attempt to address this potential confound by controlling (i.e., statistically adjusting for) state-level Republicanism. And, in fact, adding this control variable to the model looking at transmission rates does in fact "knock out" their proposed effects, yielding an association between voting restrictions and transmission rates that is no longer statistically significant.

Importantly, the authors DO still find an association between voting restrictions and death rates, even after adjusting for state-level Republicanism. However, this state-level measure is much less fine-grained than the county-level measures that constitute their primary IV (county-level voting restrictions; specifically, the Cost of Voting Index) and DVs (covid transmission and death rates). As a result, there is greater imprecision in their control variable, which could potentially lead to a kind of "type 2" error here -- that is, a failure to detect a real confounding influence of area Republicanism. Fortunately, there is an easy fix for this: use a county-level index of area Republicanism. The easiest option would be to simply look at Republican-versus-Democratic voting at the county level (e.g., for the previous two elections, akin to the Cook Partisan Voting Index). This information is readily available -- e.g., here: https://electionlab.mit.edu/data.

If the primary effects of interest hold up when controlling for this more fine-grained measure of Republicanism, I (and, I think, readers) would find the results more compelling.

A couple of other smaller comments and concerns:

In addition to the need for a county-level measure of Republicanism (which I really think is critical), I had a second concern about the authors' measure of Republicanism: I think that the Cook partisan voting index isn't ideal, insofar as it measures an area's degree of Republicanism relative to the national average in the two previous elections. Given the two-party, "us versus them" nature of American politics, though--and the way that COVID was politicized specifically along party lines--I don't think that it's RELATIVE Republicanism that is of interest, but simply a more direct measure of an area's absolute position on the Republican-Democratic spectrum. (Statistically, things will probably look very similar, but conceptually I think that this measure will make more sense to readers.)

I would also like to see what happens to the interaction effects (i.e., the interaction between voting restrictions and (1) proportion of Black residents and (2) median income) when the authors also control for the INTERACTION between voting restrictions and county-level Republicanism. This is the more stringent test of their proposed interaction, and I think that it's warranted given possible associations between county level Republicanism and both median income and racial diversity.

As a psychologist by training, I also want more insight into the mechanism behind these associations. In other words, WHY do voting restrictions relate to covid transmission and death rates? The authors speculate as to a few potential mechanisms, but do not actually test any. Of course, I recognize that this is an inherent limitation of the datasets they used, but I would suggest that the authors acknowledge this limitation a bit more clearly, and perhaps also suggest some alternative possible explanations for this association (in addition to their preferred/proposed explanation).

6. PLOS authors have the option to publish the peer review history of their article (what does this mean?). If published, this will include your full peer review and any attached files.

Reviewer #1: No

Reviewer #2: No

Reviewer #3: No

---

## [Editor Report · Decision Letter 1]

14 Apr 2022

The Relationship Between Voting Restrictions and COVID-19 Case and Mortality Rates between US Counties

PONE-D-21-34873R1

Dear Dr. Pabayo,

We’re pleased to inform you that your manuscript has been judged scientifically suitable for publication and will be formally accepted for publication once it meets all outstanding technical requirements.

Kind regards,

Natalie J. Shook

Academic Editor

PLOS ONE
---

## [Editor Report · Acceptance letter]

4 May 2022

PONE-D-21-34873R1 

The Relationship Between Voting Restrictions and COVID-19 Case and Mortality Rates between US Counties 

Dear Dr. Pabayo:

I'm pleased to inform you that your manuscript has been deemed suitable for publication in PLOS ONE. Congratulations! Your manuscript is now with our production department. 

Kind regards, 

on behalf of

Dr. Natalie J. Shook 

Academic Editor

PLOS ONE